# Wavelength-Dependent Optical Nonlinear Absorption of Au-Ag Nanoparticles

**Jun Wang** [1,2], **Yabin Shao** [1,3], **Chunyu Chen** [1], **Wenzhi Wu** [1] 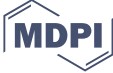, **Degui Kong** [1] and **Yachen Gao** [1,*] 

1 College of Electronic Engineering, Heilongjiang University, Harbin 150080, China; wangjun1003@163.com (J.W.); shao_yabin@163.com (Y.S.); c_cy2009@163.com (C.C.); wuwenzhi@hlju.edu.cn (W.W.); kongdegui@hlju.edu.cn (D.K.)
2 College of Light Industry, Harbin University of Commerce, Harbin 150028, China
3 Department of Computer & Electrical Engineering, East University of Heilongjiang, Harbin 150086, China
* Correspondence: gaoyachen@hlju.edu.cn; Tel.: +86-136-5454-0419

**Abstract:** The nonlinear optical absorption properties of Au-Ag nanoparticles (NPs) were studied using an open-aperture Z-scan under a nanosecond pulsed laser with wavelengths of 450 nm, 510 nm, 550 nm, and 600 nm. The experimental results demonstrated that, when the laser intensity was $1.04 \times 10^{13}$ W/m$^2$, the Au-Ag NPs showed saturated absorption (SA). When the laser intensity was increased to $3.03 \times 10^{13}$ W/m$^2$, the switch from SA to reverse saturation absorption (RSA) occurred. The nonlinear absorption and its transformation were analyzed by using local surface plasmon resonance (LSPR) effect, bleaching of ground state plasmon, and free carrier absorption theory.

**Keywords:** Au-Ag NPs; saturated absorption; reverse saturation absorption; FDTD

## 1. Introduction

The surface plasmon resonance (SPR) effect can cause metal NPs to have enhanced nonlinearity [1], which may be suitable for many prospective applications such as biosensors [2,3], biomedical applications [4,5], optical switching, optical limiting, nanophotonics [6], three-dimensional imaging [7], and antibacterial agents [8]. Among noble metal NPs, silver (Ag) and gold (Au) are extremely attractive because their SPR bands (~410 nm for Ag [9] and~520 nm for Au [10]) are tunable in visible region. The optical nonlinearity of Ag and Au NPs with different geometries has become a hot research topic in recent years, including nanotriangles [11–14], nanospheres (NSs) [15,16], nanocages [17,18], nanorods [19–23], core-shell NPs [24–26], nanodiscs [27], and nanotubes [28]. Among all nanostructures, the core-shell nanostructure is a new type of nanostructure with particular properties that are entirely different from the single-component nanostructure, which were widely used in many fields [29]. It has been reported that the core-shell structure has stronger optical nonlinearity than single-component structure [30–33]. In 2012, Y. Choi et al. synthesized hollow Au-Ag NPs and investigated their SPR [34]. In 2015, A. Monga and B. Pal investigated the surface electro-kinetics and catalytic activity of bimetallic Au-Ag core-shell nanostructures [35]. In 2016, S. Shim et al. prepared SiO$_2$-Au-Ag NSs and investigated their optical properties [36]. In 2018, A. Sakthisabarimoorthi et al. synthesized Au-Ag NPs and studied their optical nonlinearity using a Z-scan technique [26]. In 2019, S. Lin et al. successfully prepared Au-Ag NSs and evaluated their Surface-Enhanced Raman Scattering (SERS) performance [37]. In 2020, H. Lv et al. successfully synthesized plasmonic mesoporous Au-Ag NPs, and studied their electrocatalytic performance [38]. However, most of the known investigations on the optical nonlinearity of Au-Ag NPs were conducted under a single laser wavelength. In fact, the influence of excitation wavelength on the optical nonlinearity cannot be ignored. Therefore, it is of great significance to investigate the optical nonlinearity of NPs by using different laser wavelengths.

Of all the nonlinear optical properties, saturated absorption (SA) and reverse saturation absorption (RSA) have important research values. The phenomenon that the

absorption of light decreases with increasing light intensity is called SA, with which property the material can be used for laser mode locking. The phenomenon that the absorption of light increases with increasing light intensity is called RSA, with which property the material can be used for optical limiting.

In this paper, the nonlinear optical absorption properties of Au-Ag NPs were researched using open-aperture (OA) Z-scan [39,40] under 450 nm, 510 nm, 550 nm, and 600 nm nanosecond laser pulses. The saturation intensity and nonlinear absorption coefficient of materials were measured.

## 2. Sample and Experiments

The water-soluble dispersion of Au-Ag NPs was obtained from Xiamen Luman Tech Co. Ltd., Xiamen City, Fujian Province, China, with a concentration of 0.5 mg/mL and a linear transmittance of 42%. The microstructure of the Au-Ag NPs was observed using transmission electron microscope (TEM). The linear absorption spectra of the Au-Ag NPs were measured by UV-VIS spectrophotometer. The theoretical fit of liner absorption was conducted by the finite difference time domain (FDTD) method.

The nonlinear absorption properties of the Au-Ag NPs were studied by OA Z-scan experiments using a 6 ns pulsed laser (Nd:YAG). In order to produce a tunable wavelength, an optical parametric oscillator (OPO) was also used. A 2 mm-thick cuvette was used to fill the dispersion of Au-Ag NPs.

## 3. Results and Discussions

In the TEM image shown in Figure 1a the structure composed of an Au core and an Ag shell can be seen clearly, and the Au-Ag NPs have a uniform size distribution. The average diameter of the Au-Ag NPs was around 75 nm, and the Au core radius was about 26.5 nm, with a thickness of about 11 nm for the Ag shell.

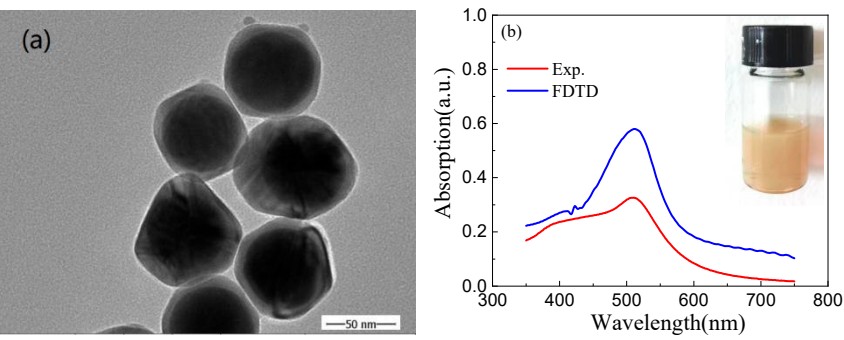

**Figure 1.** Characterization of the Au-Ag NPs. (**a**) TEM image. (**b**) Linear absorption spectra; inset image: photograph of the Au-Ag NPs' water-soluble dispersion.

As shown in the photograph in Figure 1b, the Au-Ag NPs were light brown. This resulted from the selective absorption of incident light by the Au-Ag NPs. Figure 1b displays the linear absorption spectra (red solid line) of the Au-Ag NPs. We can observe an absorption peak at 510 nm, which was from the LSPR of the Au cores. The LSPR was from the collective oscillation of conduction electrons near the Fermi level caused by an electromagnetic field, which strongly depends on composition, shape, and size of materials [41]. The blue line in Figure 1b is the simulated absorption data of a single Au-Ag NP. It can be found that the simulation data was basically similar to the experimental absorption spectra.

To verify the origin of the spectra differences, we studied theoretically the effect of the Au core radius and the Ag shell thickness on LSPR absorption spectra of Au-Ag NPs by using FDTD (the absorption mentioned here is truly exclusively the absorption, because it did not include scattering). The simulation absorption spectra are shown in Figure 2.

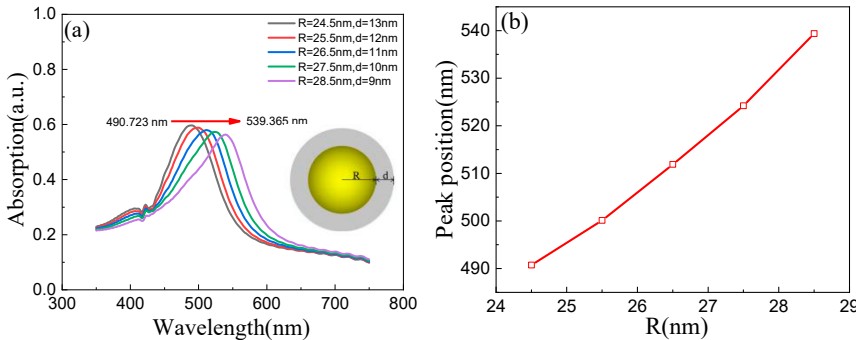

**Figure 2.** Simulation results for a single Au-Ag NP with R = 24.5–28.5 nm and d = 9–13 nm. (**a**) Absorption spectra of a single Au-Ag NP; inset image: schematic diagram of Au-Ag NPs. (**b**) Plot of peak position versus Au core radius.

We can observe from Figure 2a that the absorption peak showed an obvious redshift with an increasing radius of the Au core, when the overall sizes of Au-Ag NPs were unchanged. The change of radius of the Au core caused the absorption peak to shift. With the radius of Au core increasing, the optical properties of Au-Ag NPs were mainly characterized by the LSPR of the Au cores. Figure 2b shows the relationship between the position of the absorption peak and the radius of the Au core; we found that they were approximately linear. After the simulation, we found that the absorption peak position of the Au-Ag NPs was related to the Au core radius and the Ag shell thickness. The Au-Ag NPs we used cannot be synthesized with the same Au core radius and Ag shell thickness, and there must be a certain size distribution, so the simulation absorption spectra were not completely consistent with the experimental ones.

The repetition rate of the laser used in the OA Z-scan experiments was 10 Hz, which was low enough to avoid the thermal effect [42,43]. Previous reports generally focused on the nonlinear characteristics of the materials excited by a single laser wavelength near the resonance absorption peak. To research the nonlinear optical absorption of the samples being excited by different wavelengths, 450 nm, 510 nm, 550 nm, and 600 nm pulsed laser produced by OPO were used. Laser pulse energies of 260 μJ and 760 μJ (intensity $I_0$ at the focus of $1.04 \times 10^{13}$ and $3.03 \times 10^{13}$ W/m$^2$, respectively) were chosen for the experiments. The normalized OA Z-scan transmittance of the Au-Ag NPs is presented in Figure 3a–d.

The transmittance in this experiment was calculated by dividing the energy through the sample by that of the input sample. When the input energy was constant, the light intensity and nonlinear absorption coefficient of the sample at different Z positions was different, which made the energy passing through the sample change. Dividing the output energy of each Z point by the input energy gives the transmittance of that point. For the convenience of calculation and comparison, the normalized transmittance was obtained by dividing the transmittance of each point by the linear transmittance measured at the corresponding wavelength.

Figure 3a–d illustrate that, under the excitation energy of 260 μJ, when the sample approached the focal point, the normalized transmittance increased gradually. This phenomenon indicates that the sample exhibited SA. It can be seen that the normalized transmittance of the Au-Ag NPs at the focus was the smallest at 600 nm, and the largest at 510 nm. Under the excitation energy of 760 μJ, when the sample cuvette moved gradually to the focus, the normalized transmittance increased first and then decreased, which indicates that the sample exhibited SA at first, and then switched to RSA. Meanwhile, it can also be observed that the amplitude of RSA was the largest at 510 nm and the smallest at 600 nm.

The mechanism is explained as follows. SA results from the ground-state bleaching of the plasma absorption band. Under the excitation of a certain energy range, almost all the electrons in the ground state are pumped to the excited state, which greatly weakens the absorption capacity of NPs to external photons. Specifically, the SA was larger near the resonance absorption peak (510 nm) of the sample, while it was relatively small away

from the absorption peak. We speculate that this was related to the LSPR. With the further increasing laser energy, free carrier absorption leads to RSA dominating [44].

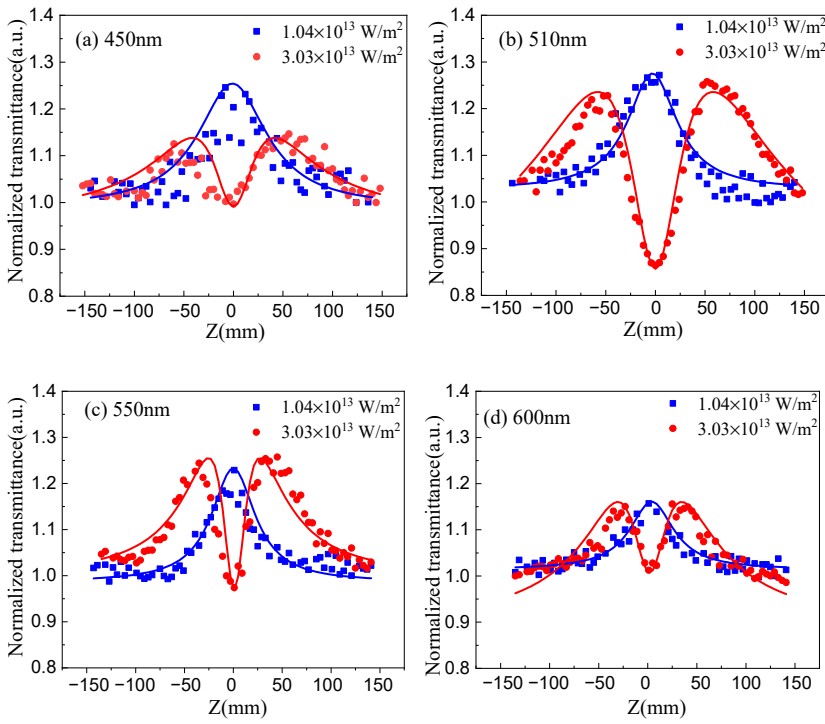

**Figure 3.** The normalized open-aperture (OA) Z-scan transmittance of the Au-Ag NPs obtained at wavelengths of (**a**) 450 nm, (**b**) 510 nm, (**c**) 550 nm, (**d**) 600 nm. The red and blue dots are experimental data, and the red and blue solid lines are theoretical one.

According to the above results, we can conclude that the Au-Ag NPs had two opposite nonlinear absorptions. The total absorption coefficient can be described as [45]:

$$\alpha(I) = \frac{\alpha_0}{1 + (I/I_S)} + \beta I \tag{1}$$

where $\alpha_0$, $I$, $I_S$, and $\beta$ are the linear absorption coefficient, laser intensity, saturation intensity, and nonlinear absorption coefficient, respectively. The laser intensity $I$ can be defined as:

$$I = \frac{I_0}{1 + z^2/z_0^2} \tag{2}$$

where $I_0$ expresses the laser intensity at the focus. So Equation (1) can be further expressed as:

$$\alpha(I_0) = \frac{\alpha_0}{1 + \frac{I_0}{(1+z^2/z_0^2)I_S}} + \frac{\beta I_0}{1 + z^2/z_0^2} \tag{3}$$

The normalized transmittance can be defined as [40]:

$$T(z) = \sum_{m=0}^{\infty} \frac{\left[\frac{-\beta I_0 L_{eff}}{1+z^2/z_0^2}\right]^m}{(m+1)^{3/2}} \tag{4}$$

where $L_{eff} = (1 - e^{-\alpha_0 l})/\alpha_0$, $L_{eff}$ expresses the effective length. Therefore, the theoretical fitting of the experimental data can be carried out by Equation (4). Figure 2 indicates that the theoretical data agreed well with the experimental data. $I_S$ and $\beta$ could be obtained by theoretical fitting, and the results are listed in Table 1.

**Table 1.** Nonlinear optical parameters of the Au-Ag NPs.

| Wavelength (nm) | E (μJ) | $I_0$ (W/m$^2$) | $I_S$ (W/m$^2$) | $\beta$ (m/W) |
|---|---|---|---|---|
| 450 | 260 | $1.04 \times 10^{13}$ | $4.32 \times 10^{11}$ | 0 |
|     | 760 | $3.03 \times 10^{13}$ | $2.16 \times 10^{12}$ | $0.79 \times 10^{-11}$ |
| 510 | 260 | $1.04 \times 10^{13}$ | $4.15 \times 10^{11}$ | 0 |
|     | 760 | $3.03 \times 10^{13}$ | $2.02 \times 10^{12}$ | $1.3 \times 10^{-11}$ |
| 550 | 260 | $1.04 \times 10^{13}$ | $4.51 \times 10^{11}$ | 0 |
|     | 760 | $3.03 \times 10^{13}$ | $2.33 \times 10^{12}$ | $1.24 \times 10^{-11}$ |
| 600 | 260 | $1.04 \times 10^{13}$ | $5.46 \times 10^{11}$ | 0 |
|     | 760 | $3.03 \times 10^{13}$ | $3.06 \times 10^{12}$ | $0.94 \times 10^{-11}$ |

Figure 4 shows the point-fold line charts of wavelength versus saturated absorption intensity, as well as the dependence of wavelength versus nonlinear absorption coefficient, which can intuitively reflect the dependence of saturation intensity and nonlinear absorption coefficient on wavelength. As a comparison, the linear absorption spectra of the Au-Ag NPs are also provided by the black solid line.

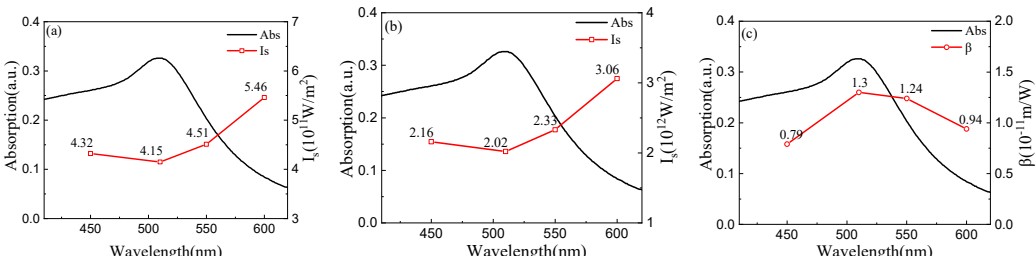

**Figure 4.** Dependence of saturation intensity and nonlinear absorption coefficient on wavelength. (**a**) Relationship between wavelength and saturated absorption intensity under 260 μJ energy excitation. (**b**) Relationship between wavelength and saturated absorption intensity under 760 μJ energy excitation. (**c**) Relationship between wavelength and saturated nonlinear absorption coefficient $\beta$.

Figure 4a,b show that the SA intensity decreased significantly when the laser wavelength was around the LSPR absorption peak. Figure 4c shows that $\beta$ increased significantly when the laser wavelength approached the LSPR absorption peak. We believe that the decrease of SA intensity around the LSPR absorption peak was due to the resonance between the incident photon and a free electron. The increase of $\beta$ near the LSPR absorption peak was caused by the enhancement of free carrier absorption.

In the Z-scan, the laser we used works at a low repetition frequency to avoid heat-induced scattering, but other kinds of scattering could not be avoid. Compared with nonlinear absorption, the scattering was weaker. However, taking into account the large size of the particles (>70 nm), the effect of scattering should not be neglected in simulation process. In fact, the optical properties of NPs are very complex. In future work, we will use a new experiment [46] to study other effects such as scattering, so as to systematically and comprehensively understand their optical properties.

## 4. Conclusions

In summary, the nonlinear optical absorption of Au-Ag NPs was studied using an OA Z-scan under 450 nm, 510 nm, 550 nm and 600 nm nanosecond pulsed laser bursts. The experimental results demonstrated that the nonlinear absorption properties of the Au-Ag NPs were not only intensity-dependent, but also wavelength-dependent. Under the excitation energy of 260 μJ, the Au-Ag NPs showed SA. Under the excitation energy of 760 μJ, the switch from SA to RSA occurred. Both the SA and RSA were strong near the resonance absorption peak (510nm) of the Au-Ag NPs, but relatively weak far away from the absorption peak.

**Author Contributions:** Methodology and writing—original draft preparation, J.W.; investigation, C.C.; formal analysis, Y.S.; data curation, W.W. and D.K.; supervision, Y.G. All authors have read and agreed to the published version of the manuscript.

**Funding:** This study was supported by the natural science fund of Heilongjiang Province (F2018027, LH2020F041).

**Institutional Review Board Statement:** Not applicable to studies not involving humans or animals.

**Informed Consent Statement:** Not applicable to studies not involving humans.

**Data Availability Statement:** Data supporting the results of this work can be obtained from the corresponding author upon reasonable request.

**Conflicts of Interest:** The authors declare no conflict of interest.

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
