# Peer review of "Wavelength-Dependent Optical Nonlinear Absorption of Au-Ag Nanoparticles"

_applsci, doi:10.3390/app11073072_

Round 1
Reviewer 1 Report
Jun Wang and coauthors report on the nonlinear absorption of bimetallic colloids. For this purpose, commercially acquired Au@Ag nanoparticles were studied by open-aperture z-scan measurements. A change from saturated absorption (SA) to reverse saturated absorption (RSA) as a function of wavelength and laser power was observed. Since the investigated bimetallic colloids were not self-produced, FDTD simulations were used to reproduce their optical signature. The parameters gold core size and silver shell thickness are not clearly known. The FDTD results do not seem to be fully traceable and it must be assumed that the simulation model used is subject to error. In fact, it seems that a lattice of particles in periodic distances (L between 150-250 nm) has been assumed, which is clearly not appropriate for a colloidal dispersion. In addition, it is well known that the optical properties of colloids of this size (approx. 70-80 nm) have non-negligible scattering contributions. However, in the entire manuscript, only absorption is mentioned and scattering contributions are not considered.
For these reasons, I recommend extensive revisions before an acceptance of this work for publication can be recommended. As an aid, I list below a series of open questions and comments to guide the revision. In principle, I consider the topic of this article to be suitable for the journal Applied Sciences and hope that the authors will show the enthusiasm to correct the existing inconsistencies.
Comments/questions:
- The authors should consider changing the title, since from my point of view, the colloids studied are not of spherical shape. They could possibly be called quasi-spherical, but the generic term "nanoparticles" and “NPs” would probably be most appropriate. It is known that Ag tends to yield angular morphologies (cubes, cuboids, etc.) using Au spheres as seeds. Spherical Au@Ag can be prepared but this involves an additional etching step using cubes as intermediate morphology (DOI: 10.1021/acs.chemmater.8b05220). Thus, I recommend to remove the term “nanospheres” and “NSs” (title, abstract, abstract and the main text) to avoid misunderstandings.
- The phenomena saturated absorption (SA) to reverse saturated absorption (RSA) and their importance for various optical applications should be explained in detail in the introduction. I anticipate that some of the readers will otherwise run the risk of not being able to understand the scientific relevance of the results presented.
- Please clarify if the “absorption” mentioned throughout the manuscript is truly exclusively the absorption or whether it is the absorbance (also known as extinction), which includes the scattering of light. This needs to be clarified throughout the whole manuscript.
- Figure 1b compared the experimental optical properties with the FDTD simulated spectrum. The match is poor, which could be a first indication that the FDTD model could be flawed. In addition, the simulation results shown in Figure 2 are hard to comprehend. As an example, it is very unlikely that an Au@Ag particle with Au core of 26.5 nm radius and an Ag shell of 4 nm shown an LSPR at 655 nm in water. I suspect that a model with a periodic arrangement of NPs was used (see parameter L), which is not appropriate. This needs to clarified. I propose to remove Figure 2b and to correct the data shown in Figure 2a. Apart from this, the insets are hardly legible due to their small sizes. It might be better to show them in full size.
- For those readers who are not familiar with OA z-scan measurements, it would be appropriate to explain how the data was normalized and how a transmittance greater than 1 can occur. Usually, transmittance is defined between 0 and 1.
- Is it possible to recreate the effects of SA and RSA using FDTD simulations? If yes, how? If not, why not?
- The authors indicate that “the nonlinear absorption and its transformation are analyzed by using […] surface plasmon resonance effect (SPR)“. This is not clear at all. First, nanoparticles do not exhibit an SPR but a localized surface plasmon resonance (LSPR). Also, how was it used to analyze the optical properties. This sounds rather wrong to my ears.
- Experimental possibilities to study the complex optical properties of nanoparticles should be briefly discussed. For example, diffuse-reflectance UV/vis/NIR spectroscopy (using an integrating sphere) allows the absorption and scattering contributions of a colloidal dispersion to be measured directly. The latter has already been successfully applied to plasmonic nanoparticles and more complex superstructures (for methodological details see DOI: 10.1021/acsami.0c16398). By distinguishing these contributions, it should be possible to describe the light-matter interaction of bimetallic colloids more accurately.
- Line 59: minor typo, “ Au NPs” should be “Au@Ag NPs”
Reviewer 2 Report
In the presented work, the influence of the laser radiation wavelength and pulse energy on the mechanism of nonlinear absorption of an aqueous colloid of shell Au @ Ag nanoparticles is studied. Using FDTF, it was shown that the linear absorption spectrum depends on the thickness of the Ag layer, with an increase in which the spectrum shifts to the red region, as well as on the average distance between the centers of the particles. It is shown that the nonlinear absorption of a colloidal solution depends on the laser radiation wavelength and pulse energy. It is shown that the effects of absorption saturation and inverse absorption saturation appear for different laser pulse energies, and their amplitude varies for different wavelengths. The authors attribute the observed effect to surface plasmon resonance on Au @ Ag nanoparticles. In general, the work was done at a high level, high-quality illustrations are provided, and the key results of the work are formalized. Authors may be interested in work 10.1021/acs.jpcb.8b11087 Comment
Replace uJ with μJ.
Author Response
Thank you fou your suggestion. I have replaced uJ with μJ.
Round 2
Reviewer 1 Report
Review 182 Appl Sci
The authors have revised their manuscript and with many corrections I am thoroughly satisfied. This includes the revised title and abstract, as well as minor text changes. The explanation of the meaning of SA and RSA is very well done.
However, not all points raised in my report were considered with sufficient care. This concerns mainly two important issues: (1) the applied simulation model of a periodic structure of NPs (with regular spacings of 300 nm) is not adequate to describe the optical properties of a colloidal dispersion (of randomly distributed NPs); (2) the assumption that the OA Z-scan measurements only measure the absorption (without scattering) is not correct. Both issues are critical to scientific soundness and must be addressed before a publication can be recommended. See details below.
Comments:
- This comment is in reference to my concerns raised in comment 4 in the first report. In a colloidal dispersion (i.e. in liquid state) the NPs are randomly distributed in the volume and thus the distances between them are randomly distributed as well. However, the authors used a simulation model that assumes a periodic arrangement with a period of 300 nm. This is clearly not appropriate and not reasonable. Also, they state that “in the sample, the center distance between Au@Ag NPs is not the same, so the simulation absorption spectra are not consistent completely with the experimental one“, which seems like a poor excuse for using an inappropriate model. I strongly discourage the authors use a periodic structure in their simulations.
- This comment is in reference to my concerns raised in comment 3 in the first report. The authors replied that “The “absorption” mentioned in OA Z-scan part is also truly the absorption” and this was justified by indicating that “Z-scan experiment was conducted with low repetition rate (10 Hz) laser to reduce and avoid thermal accumulation and scattering.” This description seems to be incorrect. To clarify, the excited plasmonic mode can decay by two different decay channels, either non-radiatively or radiatively. The non-radiative decay involves mainly the relaxation by generation of heat and high-energy carriers (e.g. hot electrons). The radiative decay describes the relaxation by reemission of a photon, this is the origin of diffuse scattering. Here, the repetition rate does not matter because the plasmon decay channels are fundamental processes. It is true that thermal accumulation can be avoided by short pulses at fast repetition but scattering cannot. Please clarify. I suppose the authors wanted to say that they assume that the scattering contributions are low (compared to the absorptive contributions), however, this is not mentioned, and taking into account the large size of the particles (>70 nm), scatting can and should not be neglected. In fact, even by the naked eye this can be witnessed, as can be seen in the photograph of the colloidal dispersion in Figure 1b. The liquid looks a bit turbid and hazy; this is the diffuse scattering of light. Consequently, this also explains why the red and blue lines in Fig.1b is a match very poorly. The authors do not measure absorption but a combination of absorption and scattering in all their experiments. This needs to be clarified.
- This comment is in reference to my concerns raised in comment 5 in the first report. The authors failed to explain how the transmittance data was normalized. Please explain in the manuscript. How was the “normalized transmittance” defined and calculated from the intensity I(z)? Please clarify for the readership unfamiliar with OA Z-scan measurements.
